# Designing and Fabricating Ordered Mesoporous Metal Oxides for CO_2_ Catalytic Conversion: A Review and Prospect

**DOI:** 10.3390/ma12020276

**Published:** 2019-01-16

**Authors:** Yan Cui, Xinbo Lian, Leilei Xu, Mindong Chen, Bo Yang, Cai-e Wu, Wenjing Li, Bingbo Huang, Xun Hu

**Affiliations:** 1Collaborative Innovation Center of the Atmospheric Environment and Equipment Technology, School of Environmental Science and Engineering, Nanjing University of Information Science & Technology, Jiangsu Key Laboratory of Atmospheric Environment Monitoring and Pollution Control, Nanjing 210044, China; cy199311@sina.com (Y.C.); lianxinbo@sina.com (X.L.); yangbo1987@nuist.edu.cn (B.Y.); LWJ_NUIST@163.com (W.L.); 13182805780@163.com (B.H.); 2College of Light Industry and Food Engineering, Nanjing Forestry University, Nanjing 210037, China; wucaie@njfu.edu.cn; 3School of Material Science and Engineering, University of Jinan, Jinan 250022, China

**Keywords:** design and fabrication strategy, ordered mesoporous metal oxides, heterogeneous catalyst, carbon dioxide, catalytic conversion

## Abstract

In the past two decades, great progress has been made in the aspects of fabrication and application of ordered mesoporous metal oxides. Ordered mesoporous metal oxides have attracted more and more attention due to their large surface areas and pore volumes, unblocked pore structure, and good thermal stabilities. Compared with non-porous metal oxides, the most prominent feature is their ability to interact with molecules not only on their outer surface but also on the large internal surfaces of the material, providing more accessible active sites for the reactants. This review carefully describes the characteristics, classification and synthesis of ordered mesoporous metal oxides in detail. Besides, it also summarizes the catalytic application of ordered mesoporous metal oxides in the field of carbon dioxide conversion and resource utilization, which provides prospective viewpoints to reduce the emission of greenhouse gas and the inhibition of global warming. Although the scope of current review is mainly limited to the ordered mesoporous metal oxides and their application in the field of CO_2_ catalytic conversion via heterogeneous catalysis processes, we believe that it will provide new insights and viewpoints to the further development of heterogeneous catalytic materials.

## 1. Introduction

The porous materials have been widely investigated and applied in many fields owing to their outstanding structural properties. According to the definition of International Union of Pure and Applied Chemistry (IUPAC), porous materials can be categorized into three types: microporous materials (pore size < 2 nm), mesoporous materials (2–50 nm) and macroporous materials (pore size > 50 nm) [1]. The most famous member of the family of microporous materials is zeolite, which has narrow and uniform pore size distribution due to its crystallographic pore system. However, zeolites exhibit serious mass transfer limitations when involving large reactant molecules. So far, attempts to improve the diffusion of reactants to catalytic centers have mainly focused on reducing the size of zeolite crystals, increasing the pore size of zeolite, and constructing additional mesoporous systems in microporous crystals [2,3,4,5]. An important research topic is to expand the pore size to the mesoporous range, allowing larger molecules to enter the porous system [6,7]. 

The development of mesoporous materials in nanometer range is beneficial to industrial process. Meanwhile, the uniform and adjustable mesopores provide enough monodisperse pore space for macromolecules, breaking through the size limitation of traditional microporous materials, and have advantages in catalysis, adsorption, separation, and drug and Deoxyribose Nucleic Acid (DNA) transfer [8,9,10]. In the early 1990s, Japanese scientists and Mobil’s scientists invented ordered mesoporous silica materials, respectively, which aroused great interest in this field [1,11,12]. Ordered mesoporous materials with adjustable surface area, different pore types, uniform nano-skeleton, abundant composition, periodically arranged monodisperse mesoporous space show great potential in many aspects [13]. Compared with zeolite, ordered mesoporous materials can be directly used as catalysts and promise better catalytic activity. In the Section 3.1 and Section 3.2, the syntheses of ordered mesoporous metal oxides by soft template method and hard template method will be reviewed in detail. Many researchers employed hard template method to fabricate ordered mesoporous carbon modified by lanthanide, ordered mesoporous Co_3_O_4_ and iron-nitrogen co-doped ordered mesoporous carbon-silicon nanocomposite (Si-Fe/NOMC) [14,15,16]. Xiao et al. [17] synthesized ordered mesoporous carbon materials from sugars by a modified soft template method. Han et al. [18] synthesized ordered mesoporous WO_3_/ZnO (OM-WO_3_/ZnO) using the soft template method. Compared with ordered mesoporous non-metal oxide materials, the ordered mesoporous metal oxides are widely investigated in the field of energy conversion and storage, catalysis, sensing, adsorption and separation due to high specific surface area and ordered pore structure. Therefore, the design and fabrication of the ordered mesoporous metal oxides have been the research focus. 

Climate change is considered to be one of the greatest environmental threats of our current globe [19]. Continuously increased concentrations of greenhouse gases in the atmosphere are widely believed to be the main driver of current climate change in the form of global warming [20]. The carbon dioxide capture and utilization (CCU) and carbon dioxide capture and storage (CCS) have the same goals but CCU also provides additional economic benefits that can be used to offset the cost of CO_2_ capture [21].

As a typical renewable compound, CO_2_ is attractive in the manufacture of commodity chemicals, fuels, and materials since CO_2_ is an abundant, non-toxic, nonflammable, typically renewable, and easily available synthon in organic synthesis. In recent years, various processes of thermochemistry, electrochemistry, photocatalysis and biology directly convert CO_2_ into value-added chemicals such as methanol, methane, carbon monoxide and formic acid. From both the economic and environmental point of view, one of the key points to achieve the efficient conversion of CO_2_ is the effective activation of CO_2_ through the highly active catalytic system [22]. It is necessary to use a highly efficient catalyst to obtain acceptable conversion and yield [4]. In recent ten years, ordered mesoporous metal oxides have received more and more attention for the catalytic conversion of CO_2_ because of their excellent textural properties and outstanding performances of CO_2_ activation [23]. Compared with the traditional silica based mesoporous zeolite supports, the ordered mesoporous metal oxides not only could be used as the catalyst supports but also could be directly used as the catalysts for the catalytic reactions, demonstrating great superiority.

In this review, we mainly analyzed the synthesis and properties of ordered mesoporous metal oxides, providing an effective basis for their stronger stability and higher catalytic activity. The role of ordered mesoporous metal oxides in the catalytic conversion of carbon dioxide is also reviewed to reduce carbon dioxide emission into the atmosphere through various chemical processes, such as CO_2_ hydrogenation to methanol, CO_2_ reforming of methane, CO_2_ methanation, synthesis of dimethyl ether, and reverse water–gas shift. It can be used to highlight the potential trend for future development in this area. In this review, the development progresses of the ordered mesoporous metal oxides and their applications in the field of CO_2_ catalytic conversion in recent years are carefully summarized and the future development trends are also prospected.

## 2. Types of Ordered Mesoporous Materials 

Mesoporous material refers to the porous material with pore size between micropore and macropore, typically between 2 and 50 nm. According to IUPAC classifications, mesoporous materials can be ordered or disordered in nature. The pores of the ordered mesoporous material are arranged uniformly [1]. Generally, the ordered mesoporous materials can be simply classified into non-metal oxide materials and metal oxide materials.

### 2.1. Ordered Mesoporous Non-Metal Oxide Materials

The ordered mesoporous non-metal oxides can be classified into silica-based mesoporous materials and carbon-based mesoporous materials according to the texture of the skeleton. The following is a brief introduction to these two types of mesoporous materials.

Silica is the most abundant type of mesoporous material, both in structure and morphology. In 1992, the M41S (MCM for Mobil Composition of Matter) series was synthesized by Mobil Corporation using a cationic surfactant as a template, which including a two-dimensional hexagonal phase MCM-41 with a space group of p6mm and cubic MCM-48 with a space group la3d [1]. In 1993, FSM-16 (Folded Sheet Material) mesoporous silica material was synthesized by a sol-gel process using long-chain alkyl trimethyl quater as a template agent, which opened the chapter of ordered mesoporous materials [24]. Zhao et al. [25] used a block copolymer as a template to synthesize a series of ordered mesoporous silica based zeolites with tetraethyl orthosilicate (TEOS) as the silica source under acidic conditions, and its pore size was adjustable in the range of 2 nm to 30 nm. El-Safty et al. [26] synthesized a class of well-defined highly ordered mesoporous silica materials (HOM) using nonionic surfactant Brij56 (C18EO10) as template. For silica-based mesoporous materials, they do not have active centers and are greatly limited in practical applications. However, the mesoporous material has an easily doped amorphous skeleton and a modifiable inner and outer surfaces. Transmission electron microscopy (TEM) were used to analyze the morphology and particle size of HOM monolithic silica, as shown in Figure 1 [26]. Nandiyanto et al. [27] successfully prepared spherical mesoporous silica particles with adjustable pore size in the nanometer range by organic template method.

The porous carbon materials have many excellent properties such as good electrical conductivity, strong skeleton rigidity, and a large specific surface area. These characteristics make the researches on porous carbon materials enduring. The synthesis of porous carbon materials can be roughly divided into hard template method and soft template method [28,29]. Many highly ordered mesoporous resin and carbon materials were synthesized by evaporation-induced self-assembly (EISA) method. Figure 2 shows the synthesis process of ordered mesoporous carbon materials by EISA [30]. Graphitized mesoporous carbon materials combine many excellent properties and have attractive applications in electrochemical energy storage [31]. Fu et al. [32] used a hexamination and resorcinol as a carbon source to form a novel soft template method for magnetic ordered mesoporous carbon (Fe_3_O_4_-OMC) using a one-pot hydrothermal method using F127 as a template. Ryoo et al. [33] used mesoporous silica molecular sieve as template to synthesize ordered carbon molecular sieves exhibiting Bragg diffraction of X-ray lines. Such carbon replicas consist mainly of nanorods (line) arrays (especially ordered mesoporous carbons with a two-dimensional hexagonal structure) so that research works on graphitization has basically been carried out around the hard template method. Juarez et al. [34] successfully synthesized ordered mesoporous carbon (OMC) with silica/triblock copolymer/sucrose composite as raw material under sulfuric acid.

### 2.2. Ordered Mesoporous Metal Oxide Materials

Ordered mesoporous metal oxides have promising applications in many fields due to their structural regularity, adjustable pore size, and high specific surface area [13]. Up to now, many ordered mesoporous metal oxides such as Co_3_O_4_ [35,36,37], TiO_2_ [38,39], WO_3_ [40,41], Al_2_O_3_ [42,43,44], ZrO_2_ [45,46,47], CeO_2_ [48,49,50], NiO [51,52], Cr_2_O_3_ [53,54], Sm_2_O_3_ [55], In_2_O_3_ [2,56], and UO_2_ [57] have been successfully synthesized. Although some ordered mesoporous metal oxides have been successfully synthesized, the structural stability and thermal stability still demand further improvement. The recent research progresses in the field of ordered mesoporous metal oxides are carefully summarized as below. 

Antonelli et al. [58] first synthesized mesoporous titania molecular sieves by an improved sol-gel method. Because of its good thermal stability, high specific surface area, ordered pore structure, adjustable pore size within a certain range, and easy modification of the surface, mesoporous TiO_2_ can effectively enhance its functions in photocatalysis and photoelectric conversion, showing a broad application prospect in the treatment of sewage, air purification, solar cell materials, nano-material microreactors, and biological materials [59]. Zhang et al. [60] found that it is necessary to add acetylacetone during the hydrolysis process because acetylacetone as a complexing agent can inhibit the hydrolysis rate of the titanium source, thus preventing the rapid formation of precipitates and favoring the formation of ordered mesoporous structures. Hu et al. [61] used EISA method to improved sol-gel and obtained titanium dioxide samples. They found that the constant volatilization of solvents and inorganic acids also enhances the polycondensation between inorganic particles. Zhang et al. [62] synthesized N-doped ordered mesoporous titania by simple solvent evaporation induced aggregating assembly (EIAA) method. Liu et al. [39] synthesized ordered mesoporous TiO_2_ hollow microspheres with highly crystalline thin shells. Figure 3 is a TEM image of ordered mesoporous TiO_2_ prepared after calcination.

Sulfuric acid zirconia (SZ) has potential application value in the fields of hydrocracking and hydroisomerization as highly effective solid acid catalyst, but the specific surface area of traditional of SZ is low so that its catalytic efficiency is not high. Therefore, mesoporous ZrO_2_ synthesized by a surfactant template route with high specific surface area has attracted great interests of the scientists in catalytic field [63,64]. Reddy et al. [65] used Zr(SO_4_)_2_ as the zirconium source and long-chain quaternary ammonium salt surfactant as template to synthesize hexagonal or layered mesoporous zirconia according to the electrostatic mechanism of ‘S^+^X^−^I^+^’. They found that the nature of the surfactant, crystallization temperature and crystallization time are the main factors affecting the synthesis of mesoporous ZrO_2_. Knowles et al. [66] synthesized a high specific surface mesoporous ZrO_2_ in an alkaline medium (PH = 11.4–11.7) to obtain a mesoporous ZrO_2_ powder. Its interplanar spacing d is independent of the length of the hydrocarbon chain of the surfactant, and the surface spacing d of the calcined sample is approximately linear with the length of the hydrocarbon chain of the surfactant. Zelcer et al. [67] prepared ordered mesoporous ZrO_2_ films by evaporation induced self-assembly. Figure 4 is a TEM image of ordered mesoporous ZrO_2_ prepared with different surfactants after calcination at 623 K. Large ordered mesoporous domains were observed in all polymer template samples [67]. 

As a functional material, tungsten trioxide (WO_3_) has been widely used in the fields of catalysis, electrochromism, energy storage of electrode materials and microwave materials [68]. In order to expand the practical application range of tungsten trioxide as a photocatalyst, it is an effective method to construct a mesoporous structure in which a tungsten trioxide material is designed into a regular structure with a large specific surface area. Ulrike et al. [69] firstly synthesized a series of ordered mesoporous metal oxides including WO_3_. It was found that potential of hydrogen (PH) value control is the most important factor affecting the mesoporous structure of WO_3_. The optimum PH value for synthesizing the hexagonal phase structure is about 4 to 8, and when the PH value is greater than 9, the synthesized mesoporous WO_3_ has two sets of layered coexisting structures. Zhu et al. [70] attempted to synthesize WO_3_ with a pore structure by surface modification using mesoporous silica (SBA-15) as a template, but eventually only WO_3_ nanowires were obtained. Subsequently, many literatures reported the use of phosphotungstic acid as a tungsten source and mesoporous silica (KIT-6) as a hard template. The ordered mesoporous tungsten trioxide with a specific surface area and a large pore size is prepared by using tungsten trioxide to crystallize at a high temperature in the mesopores of KIT-6 and then removing the template with Hydrogen Fluoride (HF) [71,72,73]. Feng et al. [74] prepared highly ordered mesoporous WO_3_ film by employing a template-assisted peroxopolytungstic acid sol–gel method. The TEM images of WO_3_ films are presented in Figure 5.

Alumina is a well promising catalytic material widely used in petroleum, chemical and other industries. The traditional microporous alumina catalysts have the disadvantages of causing clogging of the pores due to coking in the practical application processes, thereby rapidly deactivating. Therefore, synthesizing mesoporous alumina with large pore diameter is of great significance for reducing coking on the catalyst surface, blocking pores, and prolonging catalyst life. Bagshaw et al. [75] synthesized mesoporous alumina molecular sieves with a worm-like pore structure and a specific surface area of 500 m^2^/g using an electrically neutral polyoxyethylene ether nonionic surfactant as a template and aluminum alkoxide as an aluminum source for the first time. Compared to other metal oxides, the alumina surface contains a large number of Lewis acid sites, so that the surface acidity can be regulated. The synthesis of materials is related to the pH value of the mixed gels. With different template agents, the specific surface area and average pore size of the synthesized mesoporous alumina vary greatly, which was summarized in Table 1.

The activity of alumina with different crystal forms is different, with γ-type alumina having the highest activity. It is beneficial for applications such as catalysis, adsorption and separation [77]. Yuan et al. [78] used the solvent volatilization-induced self-assembly method for the first time to prepare highly ordered mesoporous γ-Al_2_O_3_. TEM images of γ-Al_2_O_3_ are displayed in parts a and b of Figure 6 with the corresponding fast Fourier transform (FFT) patterns [78]. At high temperature, the specific surface area of mesoporous γ-Al_2_O_3_ decreases, which will affect its industrial application [79]. The structure of the alumina will change, and it will gradually transform into inactive α-Al_2_O_3_. Therefore, how to suppress its transformation and to improve the thermal stability of mesoporous alumina has been a problem that people are keen to solve. The introduction of other metals oxides is one of the most effective methods for improving the thermal stability of alumina.

## 3. Synthetic Method of Ordered Mesoporous Metal Oxides

The core of synthetic ordered mesoporous metal oxide is how to “make holes” in order at nano-scale [80]. The precursor and the template material cooperate with each other to build an ordered composite structure at a mesoscopic scale through some interaction force. In this composite structure, the target components can be converted to three-dimensionally interconnected and rigid skeletons by a certain chemical method. The template material can be removed by certain methods, thus the mesoporous material of the template structure on the nanometer scale can be obtained. According to the different template materials used, the synthesis of mesoporous metal oxides can be generally categorized into soft template method and hard template method. The technological process of synthesizing ordered mesoporous materials by soft template method and hard template method can be clearly observed from Figure 7 [81], which will be discussed in detail as the following part.

### 3.1. Soft Template

Soft template is constructed by the polymerization of surfactant molecules. During the process of studying the synthesis of mesoporous materials by the soft template method, many scientists proposed a variety of synthesis mechanisms for preparing ordered mesoporous materials using soft template methods [82]. As shown in Figure 8, the cooperative self-assembly and “true” liquid–crystal templating processes are two main strategies to effectively synthesize ordered mesostructures [83]. In 1994, the Stucky group reported for the first time the synthesis of ordered mesoporous metal oxides such as WO_3_, Nb_2_O_5_, Fe_2_O_3_ using the soft template method (cationic surfactants and anionic surfactants) [69]. Compared with the silica-based mesoporous material, the mesoporous metal oxide material has poor thermal stability because the surfactant tolerant temperature is low during hydrothermal synthesis. This results in incomplete skeleton condensation of synthetic mesoporous materials and skeleton collapse after high temperature treatment.

The degree of polycondensation of inorganic species to form a stable intermediate product could be increased after going through hydrothermal, aging and other processes. After washing, filtration, and drying, an organic–inorganic composite precursor is obtained. Further calcining or solvent extraction to remove the surfactant can give mesoporous materials [82]. 

#### 3.1.1. Type of Soft Template Agent

During the synthesis of ordered mesoporous materials, it is critical to choose suitable template agents. The type and nature of template agents have a great influence on the formation of ordered mesoporous structures. It can even change the synthetic route of the reaction system. The surfactants used in the synthesis of ordered mesoporous materials can be either cationic, anionic, or nonionic [84].

A mesoporous material is formed by charge matching between the ionic surfactant and the inorganic source. Ionic surfactants can be used for the synthesis of aqueous systems. The main synthesis mechanisms are I^−^S^+^, I^+^S^−^, I^+^X^−^S^+^, I^−^M^+^S^−^ (S = surfactant, I = inorganic species, X^−^ = intermediate anion, M^+^ = intermediate cation) [85]. Cationic surfactants include long-chain alkyl quaternary ammonium salts such as cetyltrimethylammonium chloride (CTAC) and cetyltrimethylammonium bromide (CTAB). Anionic surfactants are classified into sulfonate and sulfate ester according to their hydrophilic groups. Danumah et al. [86] prepared a silica molecular sieve with medium and large pore structure using cetyl trimethylammonium chloride/cetyl trimethylammonium hydroxide (CTMAC/OH) and emulsion particles as template. Yada et al. [87] synthesized mesoporous alumina by homogenous precipitation of urea and sodium dodecyl sulphonate. It is believed that the surfactant dodecyl sulphonic acid initially forms a lamellar mesophase [87]. The distance between layers is determined by the amount of surfactant. After the surfactant forms the lamellar mesophase, the lamellar mesophase is transformed into a hexagonal shape as the urea is further hydrolyzed by the inter-layer shrinkage and the action of Al–OH groups between the adjacent aluminum atoms. Cabrera et al. [88] studied the synthesis route of mesoporous alumina. In the aqueous phase, cetyltrimethylammonium bromide is used in combination with triethanolamine. The ratio of surfactant, water and triethanolamine can be adjusted during the synthesis to adjust the pore size between 3.3 and 6.0 nm. This method has been extended to the synthesis of other mesoporous metal oxides. This process is very effective for adjusting the aperture size, but its repeatability is poor. Vaudry et al. [89] discussed the use of anion synthesis of mesoporous alumina route, the use of stearic acid as a structure-directing agent, synthesis in ethanol, formamide, chloroform or ether medium, aluminum alkyl alkoxide as aluminum source. The specific surface area of the ordered mesoporous alumina prepared after calcination ranges from 500 to 700 m^2^/g. The pore size distribution is within 2 nm and the orderliness is relatively good [90]. Holloand et al. [91] used sodium dodecyl sulfate (SDS) as a template to synthesize aluminum phosphate mesoporous materials in two steps. Tran et al. [92] synthesized ordered mesoporous manganese oxides by soft template CTAMnO_4_. Zhao et al. [93] prepared ordered mesoporous TiO_2_ materials via a sol-gel route using sodium dodecyl benzene sulfonate (SDBS) surfactants as soft templates. 

Nonionic surfactants are structurally oriented, self-assembled, and highly adaptable. They are widely used in the synthesis of ordered mesoporous materials with ordered frameworks and pore structures. Nonionic surfactants synthesize mesoporous materials through hydrogen bonding between organic templates and inorganic sources. Non-ionic surfactants mainly consist of polyepoxides such as polyethylene oxide (PEO), polypropylene oxide (PPO). Block copolymer surfactants are mainly composed of epoxide blocks of different chain lengths. Yang et al. [94] used a block copolymer with different chain lengths such as EO_20_PO_70_EO_20_, EO_106_PO_70_EO_106_, EO_75_BO_45_ to prepare a series of mesoporous metal oxides such as Ta_2_O_5_, WO_3_ and TiO_2_. Zhao et al. [95] showed that highly ordered mesoporous silica structures of different shapes were successfully synthesized by using alkyl polyoxyethylene oligomers and polymerized block copolymers in acidic media [95]. Bagshaw et al. [96] used PPO, PEO as a surfactant to synthesize mesoporous materials such as alumina and silica. This surfactant can be used for mesoporous material synthesis in non-aqueous media [97]. Compared with ethanol, the pore size can be adjusted by the number of functional groups of the surfactant. Shan et al. [98] used triethylene glycol (TEG) as a template and prepare ordered mesoporous alumina by hydrolyzing aluminum isopropoxide. Wang et al. [99] successfully synthesized ordered mesoporous TiO_2_ with crystal walls by soft membrane method. 

### 3.2. Hard Template

The hard template method, firstly reported by Ryoo et al with the synthesis of ordered mesoporous carbon (CMK-1) [33], is considered as an important way to synthesize mesoporous metal oxides. Most holes and walls of the hard template are in the size of 2–50 nm. Therefore, the pores of the resulting materials are also located in the mesoscale range. Different with the soft template method, the hard template method uses a porous solid material with a fixed mesoscopic structure as a template, which is a process of assembly and growth of a precursor in a confined space [100]. This controlled synthesis within a rigid framework generally does not require strict control of the hydrolysis and polycondensation of the precursor. Therefore, this method is particularly suitable for the synthesis of some mesostructured materials that are difficult to synthesize in the sol-gel process. At the same time, precursors can grow and crystallize at relatively high temperatures due to the confinement of rigid channels. Therefore, the method further extends the skeleton composition of the mesoporous material. The hard template method for the synthesis of mesoporous metal oxides involves four major synthesis steps [101]. They are the synthesis of templates, the loading of precursors, the conversion of precursors, and the removal of templates [102]. These four steps have important effects on the synthesis of ordered mesoporous metal oxides. Loss of control in any procedure can lead to the failure in copying the ordered mesoscopic structure of the template. Therefore, we need to carefully consider each link and choose the best solution to synthesize the target material. Hard template selection can be considered in the following four aspects. First, the connectivity of the template mesoporous wall is the key. Second, the surface properties of the template tunnel are also important in some cases. Third, the composition of the hard template skeleton is critical for the synthesis of certain special materials. Fourth, after considering the above three conditions. According to actual demands, people can control the template’s aperture size, wall thickness, and select templates with different macro topography. Thus, a series of inverse replication materials with different structures and properties are synthesized. During the entire hard template synthesis process, the loading of the precursor is the most critical step in the replication of the ordered mesostructure.

#### 3.2.1. Mesoporous Silica as a Template

Up to date, several mesoporous silica (such as SBA-15, KIT-6, FDU-12, and SBA-16) have been used as hard templates for the synthesis of mesoporous crystals [25,103]. The ordered mesoporous metal oxides have been synthesized using those silica templates including Cr_2_O_3_ [104], Co_3_O_4_ [35,105], In_2_O_3_ [106,107], NiO [51,52,108], CeO_2_ [49,109], WO_3_ [74,110], Fe_2_O_3_ [111], Fe_3_O_4_ [112], and MnO_2_ [113,114].

The silica template can be easily removed with the etching of HF or concentrated NaOH solution. The former is generally carried out at room temperature and can be completely removed by one treatment. The latter is safer but generally requires repeated treatment at higher temperatures (353 K) to remove most of the silica. It mainly depends on the chemical stability of the target product in both solutions. The most typical example of the synthesis of mesoporous materials using mesoporous silica as a template is the synthesis of mesoporous carbon. Ryong Ryoo et al. [33] first used MCM-48 as a template to synthesize mesoporous carbon (CMK-1) using sucrose as a carbon source. Using the same method, they synthesized the hexagonal CMK-3 using SBA-15 as a template [115]. In the synthesis of mesoporous metal oxides, Laha et al. [116] successfully synthesized mesoporous cerium oxide materials with high thermal stability using mesoporous silica as a template and inorganic cerium chloride salts as precursors. In order to study the morphology and pore structure, their TEM images are shown in Figure 9. The cerium oxide exhibits a high similarity to the cubic Ia3d symmetry over a long range [116]. Compared with the synthesis of mesoporous carbon, the difficulty in synthesizing metal oxides in this way is that the inorganic precursor is more difficult to enter into the pores of mesoporous silica. As a result, the precursor’s pore occupancy is very low. Shang et al. [117] prepared ordered mesoporous CoFe_2_O_4_ by nanocasting method using mesoporous silica SBA-15 as hard-template. For the same purpose, Tian et al. [118] used microwave etched mesoporous silica as a template to synthesize a series of mesoporous metal oxides such as Co_3_O_4_, In_2_O_3_, Cr_2_O_3_, Mn_3_O_4_, CuO, NiO, and CeO_2_. The mesoporous silica treated with microwave method not only removes the surfactant but also leaves a rich hydroxyl group in the mesopores. The hydrophilic hydroxyl groups in the pores facilitate the entry of hydrophilic inorganic salt precursors. Therefore, the synthesized material has good continuity [119]. In general, highly ordered mesoporous silica is the promising hard template for obtaining highly ordered non-silica mesoporous materials.

#### 3.2.2. Mesoporous Carbon as a Template

Using mesoporous carbon as a hard template to the synthesis of mesoporous materials is also an effective method [120]. The hard template method has strong universality, and the mesoporous structure of the target material can be controlled by selecting a hard template with different structures. In addition, the mesoporous silica used in the hard templating method has high thermal stability. It can withstand high temperatures and crystallizes most of the metal oxides on its surface to obtain mesoporous metal oxides with high crystallinity [121]. The metal ions in the soft template method are sensitive to humidity during hydrolysis and polymerization, and the products are often amorphous and have poor thermal stability [122]. However, the hard template method uses sodium hydroxide or hydrofluoric acid to remove the template and the synthesized mesoporous metal oxide must have strong acid-base resistance. The preparation process is complicated, and a mesoporous metal oxide is prepared by first synthesizing a mesoporous silica hard template and then removing the template [123].

The carbon template is mainly removed by calcining in air. For target substances that are easily oxidized by air at high temperatures, carbon templates can be removed at high temperatures by oxygen [124]. Mesoporous carbon materials are generally obtained by the hard template method to replicate ordered mesoporous silica. Therefore, using mesoporous carbon as a template to the synthesis of mesoporous materials is actually a two-step nanoreplication technique. This method of producing mesoporous silica is economically unworthy. However, this provides us with a new method for the synthesis of mesoporous materials. Mesoporous carbon can be used as a hard template to synthesize new silica based mesoporous materials [125,126,127]. Kim et al. [128] synthesized mesoporous silica (HUM-1) using CMK-1 (replicated from MCM-48) as a template. The resulting mesoporous material is different from the above SBA-15 derived from CMK-3. The mesoporous structure of HUM-1 and MCM-48 is obviously different and this mesoporous silica material HUM-1 cannot be obtained through the traditional surfactant template pathway. 

## 4. Catalytic Application of Mesoporous Metal Oxides in Catalytic Conversion of CO_2_

The cumulation of carbon dioxide in the atmosphere is widely recognized as the main cause of global warming, which may pose a huge threat to human living environment and human beings. Climate change experts recommend that it should be developed and utilized as soon as possible so as to effectively manage carbon dioxide within the limits of the atmosphere. The chemical conversion of carbon dioxide into useful products and fuels, such as CH_3_OH, CO, CH_4_, and dimethyl ether (DME), is considered as an attractive CO_2_ recovery method to control its emission into the atmosphere [129,130].

Due to the large specific surface area, developed pore structure and wide pore size, ordered mesoporous metal oxides have been considered as promising catalyst candidates for the catalytic conversion of CO_2_. In recent years, many scholars have achieved a series of promising and excellent results in this field. Here we briefly review the application of mesoporous metal oxides as catalysts in the catalytic conversion of CO_2_ via heterogeneous catalysis process.

### 4.1. CO_2_ Hydrogenation to Methanol

In the past two decades, CO_2_ has been used as a substitute for CO in methanol production and CO_2_ hydrogenation to methanol has been widely recognized as an effective CO_2_ utilization method [131]. Under certain conditions, the methanol formed by the hydrogenation process of atmospheric CO_2_ is considered to be the most economical way to alleviate the greenhouse effect caused by the significant increase in CO_2_ concentration. It is mainly because that methanol is not only an important chemical intermediate to produce some chemicals such as formaldehyde and acetic acid, but also an excellent fuel due to its cleaner emissions in comparison with other fossil fuels [132].

In the CO_2_ hydrogenation process, the main reaction is the formation of methanol and the reverse water–gas-shift reaction is a side reaction [133]:

Formation of methanol:
CO_2_(g) + 3H_2_(g) → CH_3_OH(g) + H_2_O(g)   △H_298 K_ = −90.70 kJ/mol(1)

Reverse water–gas-shift reaction:CO_2_(g) + H_2_(g) → CO(g) + H_2_O(g)   △H_298 K_ = +41.1 kJ/mol(2)

The methanol production is an exothermic reaction, in which the number of reactive molecules is reduced. Therefore, the decrease in temperature and the increase in pressure favor the reaction of thermodynamic analysis. However, considering the reaction rate and the chemical inertness of CO_2_, an increase in the reaction temperature (>513 K) favors the activation of CO_2_, which in turn forms methanol. The reverse water gas shift reaction results in additional hydrogen consumption and a reduction in methanol production. The formation of a large amount of water has an inhibitory effect on the active metal, resulting in deactivation of the catalyst. Therefore, the hydrogenation of CO_2_ to methanol requires a more selective catalyst to avoid the production of unwanted by-products [134]. 

Most methods of conversion require huge amounts of energy and lengthy procedures and complex instrumentation, owing to the fact that the CO_2_ molecule is inherently very stable and inert. However, the simple, less cumbersome and cost-effective method of CO_2_ conversion based on photo-catalytic reduction has become quite attractive. In photo-catalytic CO_2_ reduction, the electron-hole pairs generated on the surface of a semiconducting photo-catalyst mediates photo-oxidation and photo-reduction reactions that result in the desired end product [135].

Gondal et al. [2] synthesized ordered mesoporous indium oxide nanocrystal (m-In_2_O_3_) by nanocasting technique, in which highly ordered mesoporous silica (SBA-15) was used as structural matrix. The results showed that the introduction of mesoporosity in indium oxide, and the consequent enhancement of positive attributes required for a photo-catalyst, transformed photo-catalytically weak indium oxide into an effective photocatalyst for the conversion of CO_2_ into methanol. Richardson et al. [136] prepared Mn and Cu doped titanium dioxide by sol-gel method and obtained different nanocomposites for CO_2_ conversion to methanol. Compared to commercial catalysts, the band gap of Mn and Cu doped TiO_2_ is less than 3 eV. Due to the rapid transport of excited state electrons to the metal dopant, the coupling between Mn and Cu is achieved, which enhances the ability of CO_2_ photocatalytic reduction to methanol. 

### 4.2. CO_2_ Reforming of Methane (CRM)

CH_4_ and CO_2_ are rich in natural resources. Therefore, it is important to convert these two molecules into high-value added compounds [137]. The reaction of CO_2_ reforming of CH_4_ to produce syngas (i.e., CO + H_2_) can be used in chemical energy transfer systems as well as in the production of liquid fuels [138]. In the absence of water, reforming can be carried out with carbon dioxide instead of water to form a syngas having a lower H_2_/CO ratio.
CO_2_(g) + CH_4_(g) → 2CO(g) + 2H_2_(g)   △H_298 K_ = +274.3 kJ/mol(3)

For the CRM reaction catalysts, most of the VIII family metal catalysts were investigated, including non-precious metal Ni and Co based catalysts and noble metal catalysts [139]. The Ni-based catalysts are considered to be potential catalysts for CRM reactions because of their low cost and high catalytic activity. Under CRM reaction conditions, the rapid deactivation of catalyst caused by thermal agglomeration and surface coke of metal active centers are major obstacles to industrialization. It was reported that the Ni clusters in small size have good abilities to inhibit coke formation [139]. The ordered mesostructures can provide the gaseous reactants with more accessible metallic active centers than traditional catalysts, thus preforming higher catalytic activity. In addition, the confinement effect of mesopores can effectively inhibit the thermal sintering of Ni nanoparticles and promise better catalytic stability [140]. In recent years, Xu et al. have carried out a series of systematic researches and successfully synthesized the ordered mesoporous metal oxides such as Co-Ni-MO (M = Mg, Ca)-Al_2_O_3_ [141], CoO-MO (M = Mg, Ca)-Al_2_O_3_ [142], xCoyNi-Al_2_O_3_ [139], Ni/-CexZry [143], Ni/CaO-Al_2_O_3_ [144] and NiO-CaO-Al_2_O_3_ [145] by EISA strategy and used as catalysts or catalytic supports for CRM reactions. TEM characterizations of the ordered mesoporous Al_2_O_3_ (OMA) further confirmed the existence of ordered mesoporous channels. The images are shown in Figure 10. Kim et al. [146] prepared mesoporous Ni-Mg_x_-Al_2_O_3_ (NMA) catalyst by modified EISA method and evaluated its catalytic performance. Since MgO has a strong Lewis basicity and promotes CO_2_ activation, the M-NMA catalyst is more resistant to carbon formation than the non-promoted catalyst. Compared with the Ni/Al_2_O_3_ catalyst (NA), the M-NMA catalyst has a larger surface area and a narrower pore size distribution [147]. The catalysts with excellent long-term stability are considered to be the most important concern in the field of CRM reactions. As shown in Figure 11a,b, these ordered mesoporous catalysts showed almost no significant deactivation of activity [141]. Besides, it is believed that the mesoporous ceria-zirconia solid solution carriers could activate CO_2_ and finally participate in the process of CRM reaction and carbon elimination via their redox properties so that it could be considered as a series of promising catalyst carriers for CRM. This typical process was expressed in Scheme 1. Xu et al. [143] prepared mesoporous nanocrystalline Ce-Zr solid solution with different Ce/Zr ratios by improving the evaporation induced self-assembly strategy. The results shown that the sample with the Ce/Zr ratio of 50/50 and the Ni loading amount of 7 wt% or 10 wt% has the highest catalytic activity.

### 4.3. CO_2_ Methanation

Natural gas is considered as a potential source of energy due to its clean nature. Therefore, the conversion of carbon dioxide to methane can not only reduce greenhouse gas emissions but also develop carbon dioxide. Compared with the production of hydrocarbons and alcohols, CO_2_ methanation is more thermodynamic favorable because of its strong exothermic properties [148].
CO_2_(g) + 4H_2_(g) → CH_4_(g) + 2H_2_O(g)   △H_298 K_ = −165.0 kJ/mol(4)

Catalytic supports have a significant effect on the high dispersion of metal active sites, which greatly promotes the activation and dissociation of H_2_ molecules. Therefore, a material having a large surface area, a large pore volume, and a clear channel such as a mesoporous metal oxide can be used as a carrier or catalyst for the CO_2_ methanation catalyst [149,150]. Compared with the traditional supported catalysts, ordered mesoporous catalysts can inhibit the sintering of active metal nanoparticles in a certain space due to their superior confinement effect [5,6]. The ordered mesoporous materials generally have relatively high surface areas, which facilitates the high dispersion of the active metal and increases the activity of the catalyst [151,152].

Previous studies on CO_2_ methanation catalysts have focused on the loading of Ru, Rh, Ni and Pd on the catalyst. However, their high prices and high hydrogenation temperature (>573 K) limit the application of precious metal catalysts in catalytic hydrogenation of CO_2_ [153,154]. The Ni-based non-precious metal catalysts have the advantages of low cost and high catalytic activity [155]. The disadvantage of Ni-based catalysts is that they are greatly inclined to sinter during the reaction. Thereby, effectively inhibiting the thermal sintering of the metal Ni nanoparticles under the conditions of CO_2_ methanation has become the challenge and research focus in this field.

Xu et al. successfully synthesized NiO-OMA [149], OMA-*x*Co*y*Ni [156], OMA-10Ni*x*Ca [157], OMA-10Ni*x*Mg [158], and OMA-10Ni3Re (Re = La, Ce, Sm and Pr) [159], which would be used in the CO_2_ methanation reaction. The conversion of CO_2_ on the OMA-10Ni and OMA-10Ni5Mg catalysts is also much higher than the 10Ni/Al_2_O_3_ catalyst, especially in the low temperature. Owing to the excellent textural properties of the ordered mesoporous catalysts, the gaseous reactants are easily to diffuse toward the accessible metallic Ni active site. The effects of reaction temperature on the catalytic activity and selectivity of OMA-10Ni, OMA-10Ni5Mg and 10Ni/Al_2_O_3_ were investigated in Figure 12 (1) and (2) [158]. Liu et al. [160] used the EISA method to prepare ordered mesoporous Ni-Co/Al_2_O_3_ (NCOMA) catalysts for CO_2_ methanation. The ordered mesoporous 10N3COMA catalyst has high activity with the maximum CO_2_ conversion rate of 78% and the CH_4_ selectivity of 99% under specific conditions (673 K, 0.1 MPa, 10,000 mL·g^−1^·h^−1^). The results showed that the Co species could significantly increase the H_2_ uptake and the catalyst showed high stability and superior anti-sintering property due to the confinement effect of the ordered mesostructure. The CO_2_ photoreduction is also considered as an effective route to produce CH_4_. Wang et al. [99] combined the evaporation-induced self-assembly process with the two-step calcination process to successfully synthesize ordered mesoporous TiO_2_ with crystalline walls. Ordered mesoporous TiO_2_ has higher CH_4_ production efficiency and better CO_2_ photoreduction stability than the disordered mesoporous counterpart. The superior performance of ordered mesoporous TiO_2_ for CO_2_ photoreduction may be due to the limited spatial effects of ordered mesoporous structures. In the ordered mesoporous structure, the mass transfer of gas molecules is more stable than that of disordered mesoporous structures.

### 4.4. Synthesis of Dimethyl Ether (DME)

DME is considered as a kind of future fuel, especially as a diesel alternative. DME has higher thermal efficiency compared to conventional fuels. Besides, it also does not emit sulfur oxides or soot. In addition, DME can be used as a raw material for a range of chemicals such as oxygenates, olefins, and hydrocarbon fuels (gasoline, aviation fuel). Therefore, DME will become a very important clean fuel in the view of sustainable development, which will help the effective management of the future energy [161]. The hydrogenation of CO_2_ or syngas can produce DME. Generally, this reaction can be divided into two steps. The first step is the synthesis of methanol from carbon dioxide or synthesis gas under the action of a catalyst. In the second step, methanol is dehydrated on the catalyst to produce DME. The following are the main reactions that occur during the synthesis of DME from CO_2_/H_2_ gas mixtures.

Methanol synthesis reaction:CO_2_(g) + 3H_2_(g) → CH_3_OH(g) + H_2_O(g)   △H_298 K_ = −49.4 kJ/mol(5)

Methanol dehydration reaction:2CH_3_OH(g) → CH_3_OCH_3_(g) + H_2_O(g)   △H_298 K_ = −23.4 kJ/mol(6)

It can be seen from the above reactions (5) and (6) that the synthesis of DME by CO_2_/H_2_ involves methanol synthesis and methanol dehydration [162,163,164].

Thus, many bifunctional (or mixed) catalytic systems contain metal sites for the hydrogenation of CO_2_ and acid sites for the continuous dehydration of alcohols to ethers, especially in the reaction of syngas directly synthesizing DME. Both the active site of the solid acid and the active site of the metallic copper are related to the catalytic activity [165,166]. Ham et al. [167] uses the evaporation induced self-assembly method (EISA) to synthesize ordered mesoporous γ-Al_2_O_3_. In the direct synthesis of DME from syngas, the catalytic activity and stability was improved because the amount of copper crystals accumulated was small. The copper nanoparticles form a strong interaction with the acidic sites on the surface of ordered mesoporous Al_2_O_3_ through the formation of CuAl_2_O_4_ interface which provides some effective acidic sites for DME methanol dehydration. The highly dispersed Cu nanoparticles have a strong interaction at the rich acidic sites of the ordered mesoporous Al_2_O_3_, which greatly improves the stability of the catalyst and the DME selectivity. The TEM image of Cu/mesoAl is displayed in the Figure 13 [167]. Dehydration of methanol is the second step of the conversion of CO_2_ to dimethyl ether. Luan et al. [168] successfully prepared high surface area mesoporous γ-Al_2_O_3_, which was used for dehydration of dimethyl ether and helped convert CO_2_ to dimethyl ether.

### 4.5. CO_2_ Reverse Water Gas Shift (RWGS) Reaction

The reverse water gas shift reaction (RWGS) converts CO_2_ to CO, which can be further converted into hydrocarbons by the Ficher–Tropsch synthesis process. The RWGS reaction is another key reaction in the field of catalytic hydrogenation of CO_2_, which has been considered as a promising candidate for large-scale conversion of CO_2_ and renewable H_2_ [169,170]. The following is the main reaction in the RWGS process.
CO_2_(g) + H_2_(g) → CO(g) + H_2_O(g)   △H_298 K_ = 41.1 kJ/mol(7)

Pt, Co, and Ni based catalysts have been widely investigated in the RWGS reaction [171,172,173]. The main problem with these catalysts is the progress of the methanation reaction, which will decrease the selectivity of the RWGS reaction. The highly dispersed and small particle size metals are the key to the preparation of highly active and highly selective RWGS catalysts [174].

CeO_2_ is a typical rare earth metal oxide with a cubic fluorite structure. Under a reducing atmosphere, the surface oxygen of CeO_2_ can be reduced and oxygen vacancies will be generated subsequently. Oxygen vacancies play a key role in the catalytic reaction, especially in the catalytic reduction of RWGS [175]. Dai et al. [176] synthesized the ordered mesoporous CeO_2_ by hard template to carry out the RWGS reaction and compared it with non-porous CeO_2_. According to the TEM results in Figure 14, the non-porous CeO_2_ (b), (c) catalysts have low porosity and small specific surface area, which is disadvantageous for absorption and activation of reactant molecules. The ordered mesoporous CeO_2_ (a) catalyst has a good ordered mesoporous structure to facilitate the adsorption and activation of CO_2_ and H_2_ molecules. Therefore, ordered mesoporous CeO_2_ (a) catalyst has a good catalytic activity in CO_2_ RWGS reaction [176].

The addition of other metal elements to the catalyst also can increase the catalytic activity of the RWGS and the selectivity of the catalytic reaction [177]. The transition metal (Ni, Co, Fe, Mn, Cu) can be dissolved in the CeO_2_ lattice to form a solid solution, generating oxygen vacancies [178]. Wang et al. [179] prepared a series of mesoporous Co-CeO_2_ catalysts with different Co contents by colloidal solution combustion method and carried out the CO_2_ RWGS reaction. In the Co-CeO_2_-M catalyst, the Co_3_O_4_ particles embedded in the pore walls were separated by fine CeO_2_ particles and strongly interacted with CeO_2_. The results showed that the 5% Co-CeO_2_-M catalyst was provided with the best catalytic performance for RWGS reaction. Dai et al. [180] prepared mesoporous M-CeO_2_ (M = Ni, Co, Fe, Mn, Cu) catalysts by hard template method and carried out CO_2_ RWGS reaction. The CO_2_ RWGS reaction performance has a great relationship with the d orbital holes of the transition metal. It can be seen from Figure 15 that on the NiCE, CoCE, FeCE, MnCE, CuCE, and CeO_2_ catalysts, the conversion rate of CO_2_ increases as the reaction temperature increases.

## 5. Conclusions and Perspective

There are many strategies for the design and preparation of ordered mesoporous catalysts. Ordered mesoporous materials have attracted wide attention due to their rich unique properties, functions and potential application prospects. In recent years, many achievements have been made in its synthesis and structural characterization. The classification of ordered mesoporous materials, the synthesis of ordered mesoporous metal oxides and their applications in catalytic conversion of carbon dioxide are reviewed. It is believed that relevant scientists can obtain information on the synthesis, properties and potential applications of these materials to facilitate their research. Ordered mesoporous metal oxide materials are important components of catalytic materials and have the unique physicochemical properties of metal oxides.

In recent years, a variety of ordered mesoporous metal oxides have been synthesized by soft and hard template methods and their properties and applications investigated. The methods and mechanisms for the synthesis of ordered mesoporous metal oxides by soft and hard template methods have matured, but both methods have advantages and disadvantages. The advantages of the soft template method are that the template cost is relatively low, the synthesis method is simple and the conditions are mild. The main disadvantages are that the metal ions in the hydrolysis and polymerization processes are relatively sensitive to moisture, and the products are often amorphous and have poor thermal stability. The advantages of the hard template method are that it is universal and the mesoporous structure of the target material can be controlled by selecting hard templates with different structures. The disadvantage of the hard template method is that when the template is removed with sodium hydroxide or hydrofluoric acid, the mesoporous metal oxide synthesized must be resistant to strong acids and bases. The preparation process is cumbersome and complex. 

Ordered mesoporous metal oxide can catalyze the conversion of carbon dioxide to value-added chemicals and fuels. As a rich natural raw material, carbon dioxide has attracted great interests in recent years. The key point to CO_2_ conversion is the activation of either CO_2_ or co-reactant at different conditions, especially at low temperature. In this way, catalytic conversion of CO_2_ has been carried out by different methodology, including CO_2_ reforming of methane to syngas production over catalysis, CO_2_ hydrogenation for methanol synthesis by mesoporous catalyst, CO_2_ methanation over a Ni based ordered mesoporous catalyst, synthesis of DME from CO_2_/H_2_ gas mixture, and CO_2_ reverse water–gas shift. In the future research, not only should the catalytic conversion of CO_2_ be further improved, but also the hydrothermal stability, chemical stability and environmental compatibility of ordered mesoporous metal oxide need to be greatly optimized. Meanwhile, for the wide application of materials in industry, the cost of materials should be minimized and it is required to create new ordered mesoporous metal oxide materials that are easy to be synthesized in large scale with low cost.

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
