# Peer review of "Designing and Fabricating Ordered Mesoporous Metal Oxides for CO2 Catalytic Conversion: A Review and Prospect"

_materials, 2019, doi:10.3390/ma12020276_

Round 1

Reviewer 1 Report

In this paper, the characteristics, classification, and synthesis of ordered mesoporous metal oxides together with the catalytic application of ordered mesoporous metal oxides in the field of carbon dioxide conversion are discussed. The paper also considers the field of CO2 catalytic conversion via heterogeneous catalysis processes with regard to the carbon emission reduction. The paper covers a wide range of studies on the mesoporous metal oxides and the scope of the paper is in line with the one of the Processes Journal. However, the paper can be published only if the following remarks are addressed carefully.

1.       Lines 51-58: these are all referred to hard template method. It is not needed to mention the name of the authors one by one; all of these references can be given at the end of a phrase for example "many researchers used hard template method to make carbon-silica etc. catalysts [references]".

2.       Line 53: Mention that the reader can refer to section 3.1 and 3.2 to find what hard and soft templated are.

3.       Lines 63-75: As the introduction started with talking about porous materials, this part seems very dissimilar and discrete from the rest of the introduction. I suggest that there is no need to explain the issue of climate change and CO2. Very shortly and in a few words jump to the main point which is the utilization of CO2.

4.       Line 83-86: give references.

5.       Lines 87-92: was this done by you? Who did that? Is this the scope of this paper? Clarify, or give references.

6.       ALL the Figures: Most of the time the texts are not readable. They are too tiny to be seen. Very unclear image compared to the original paper. They seem like a snapshot. Please acquire the original image or make a clear copy of the image. This does not look like the images in the reference. Images are nor consistent. Some look deformed and distorted and the font sizes in the figures are not consistent.

7.       Line 146: what unique properties? What are those properties?

8.       Lines 431 and 433: Use the correct mark for a balanced or reversible reaction! "=" is not the one! Do this for all the reactions in the paper. And give the reactions ordering numbers (1, 2, 3,..) like figures and tables.

Author Response

Comments : In this paper, the characteristics, classification, and synthesis of ordered mesoporous metal oxides together with the catalytic application of ordered mesoporous metal oxides in the field of carbon dioxide conversion are discussed. The paper also considers the field of CO2 catalytic conversion via heterogeneous catalysis processes with regard to the carbon emission reduction. The paper covers a wide range of studies on the mesoporous metal oxides and the scope of the paper is in line with the one of the Processes Journal. My review comments are as follows.

Answer: Thank you very much for providing such constructive and valuable suggestions to us. Your valuable comments are of great significance to improve the quality of the manuscript. Following your valuable suggestions, the authors have made the best efforts to revise the whole manuscript. Please kindly find the revised manuscript as a reference. The authors sincerely wish that the revised manuscript could make you satisfied. Any further suggestions as well as comments are also greatly welcome and appreciated.

Q1. Lines 51-58: these are all referred to hard template method. It is not needed to mention the name of the authors one by one; all of these references can be given at the end of a phrase for example "many researchers used hard template method to make carbon-silica etc. catalysts [references]"

Answer: Thanks you very much for your advisable suggestion. Following your advice, the authors have made the corresponding modification by avoiding mentioning the name of the authors one by one. The related references are summarized and provided in your suggestive style. Specifically, the authors summarized the researchers using the hard template method to make materials and provided references at the end of the phrase in a brief way. Please kindly find the revised manuscript as a reference.

Q2. Line 53: Mention that the reader can refer to section 3.1 and 3.2 to find what hard and soft templated are.

Answer: Thank you very much for your constructive comments. According to your comments, the authors have pointed out that the readers can refer to the hard and soft template method in section 3.1 and 3.2 before introducing synthetic materials in detail. Please kindly find the revised manuscript as a reference.

Q3. Lines 63-75: As the introduction started with talking about porous materials, this part seems very dissimilar and discrete from the rest of the introduction. I suggest that there is no need to explain the issue of climate change and CO2. Very shortly and in a few words jump to the main point which is the utilization of CO2.

Answer: Thanks you very much for your advisable suggestion. The authors also agree with your viewpoint that there is no need to explain the problem of climate change and carbon dioxide in such detailed manner. Therefore, this paragraph has been simplified and presented in a brief manner according to your sensible advice. Please kindly find the revised manuscript as a reference.

Q4. Line 83-86: give references.

Answer: Thank you very much for your constructive comments. The authors have modified this section and updated the related references in this part according to your comments. Please kindly find the revised manuscript as a reference.

Q5. Lines 87-92: was this done by you? Who did that? Is this the scope of this paper? Clarify, or give references.

Answer: Thanks you very much for your advisable suggestion. Following your advice, the authors have modified this part and it covers the scope of this review. The related references have been provided. Please kindly find the revised manuscript as a reference.

Q6. ALL the Figures: Most of the time the texts are not readable. They are too tiny to be seen. Very unclear image compared to the original paper. They seem like a snapshot. Please acquire the original image or make a clear copy of the image. This does not look like the images in the reference. Images are nor consistent. Some look deformed and distorted and the font sizes in the figures are not consistent.

Answer: Thank you very much for your advisable suggestion. Following your advice, the authors have modified all the figures to make them as consistent and clear as possible. The authors make sure that the images in this manuscript are originally acquired from the reference paper. Please kindly find the revised manuscript as a reference.

Q7. Line 146: what unique properties? What are those properties?

Answer: Thank you very much for your constructive questions. The authors have provided the specific properties of ordered mesoporous metal oxides in this section according to your questions. Please kindly find the revised manuscript as a reference.

Q8. Lines 431 and 433: Use the correct mark for a balanced or reversible reaction! "=" is not the one! Do this for all the reactions in the paper. And give the reactions ordering numbers (1, 2, 3,..) like figures and tables.

Answer: Thank you very much for providing such encouraging and constructive suggestions for us. Following your suggestions, the authors have revised all the reaction equations with the formula format and marked the reactions with ordering numbers like figures and tables. Please kindly find the revised manuscript as a reference.

Reviewer 2 Report

This work presents several weak points. However, from my point of view the most relevant is related to language. On one side, several parts where written without paying attention to the terms typically used in the literature for describing the same concepts. On the other side, a great number of mistakes can be found (I have accounted more than 10 in some pages). To be remarked is that when some phrases presented a lack of errors I typed them on google scholar and found them in other works. Despite English mistakes, the introduction to the topic was not clear. A clear comparison with other materials should be done to justify why to write a review on these types of materials. For instance, zeolites are referred as not interesting for this reaction due to the pore sizes while CO2 or H2 are known to present lower kinetic diameters than the pore dimensions of zeolitic materials. 

Even if the review regarding the preparation conditions, characteristics or, for instance, composition of these types of materials was correct, a very deep review regarding English and the writing style in general should be carried. Also, the hole work should be reformulated in order to justify its publication. For these reasons, I recomend the rejection of this work.

Author Response

Comments: This work presents several weak points. However, from my point of view the most relevant is related to language. On one side, several parts where written without paying attention to the terms typically used in the literature for describing the same concepts. On the other side, a great number of mistakes can be found (I have accounted more than 10 in some pages). To be remarked is that when some phrases presented a lack of errors I typed them on google scholar and found them in other works. Despite English mistakes, the introduction to the topic was not clear. A clear comparison with other materials should be done to justify why to write a review on these types of materials. For instance, zeolites are referred as not interesting for this reaction due to the pore sizes while CO2 or H2 are known to present lower kinetic diameters than the pore dimensions of zeolitic materials.

Even if the review regarding the preparation conditions, characteristics or, for instance, composition of these types of materials was correct, a very deep review regarding English and the writing style in general should be carried. Also, the whole work should be reformulated in order to justify its publication. For these reasons, I recommend the rejection of this work. My review comments are as follows.

Answer: Thank you very much for providing such constructive and valuable suggestions to us. Your valuable comments are greatly important to improve the quality of manuscript. Following your valuable comments, the authors have tried our best to revised the whole manuscript to avoid any spelling and grammar errors. The author has realized the importance of the language of the article and will strive to achieve better, which will be also paid more attention in our future manuscripts. The justification why to write a review on these types of materials had been provided in the revised manuscript. Besides, this review regarding the preparation conditions, characteristics or composition of these types of materials have been carefully checked to avoid any mistake. Please check the revised manuscript for reference. The authors sincerely wish that the revised manuscript will make you satisfied. Any further suggestions and comments are also greatly welcome and appreciated.

Reviewer 3 Report

I have reviewed the review article "Designing and Fabricating Ordered Mesoporous Metal Oxides for CO2 Catalytic Conversion: A Review & Prospect" written by Yan Cui et. al.

The articles is well organised with introduction of different pore size materials, how they are synthesised and what template methods are successful so far.  The paper also highlighted both soft and hard template methods and their application of mesoporous metal oxide synthesis.

At the end the application of these mesoporous metal oxides for the conversion of CO2 via different reactions modes such as methanol synthesis, CO2 reforming of methane, methanation, DME synthesis, RWGS was elaborately with relevant references, figures and pictures.

I enjoyed reading the article and hope the audience would also feel the same.

Author Response

Comments : The articles is well organised with introduction of different pore size materials, how they are synthesised and what template methods are successful so far. The paper also highlighted both soft and hard template methods and their application of mesoporous metal oxide synthesis. At the end the application of these mesoporous metal oxides for the conversion of CO2 via different reactions modes such as methanol synthesis, CO2 reforming of methane, methanation, DME synthesis, RWGS was elaborately with relevant references, figures and pictures. My review comments are as follows.

Answer: Thank you very much for providing such encouraging and constructive comments for us. This manuscript reviews the synthesis strategies of the ordered mesoporous metal oxides and their potential applications in the field of catalytic conversion of carbon dioxide. We have tried our best efforts to improve the quality of the manuscript and made corresponding revisions in the revised manuscript according to other two reviewers’ comments. The authors believe that this review paper can bring holistic interests to scientists in the realms of CO2 resource utilization and heterogeneously catalytic conversion. The authors sincerely wish that we can make much greater contribution in the fields of ordered mesoporous metal oxide materials and CO2 catalytic conversion in the future work.

Round 2

Reviewer 1 Report

The authors have addressed all the comments and there is no more comment or remarks to cover. The manuscript in the current form is acceptable. 

Author Response

Thanks so much for your encouraging comments. The authors appreciate this very much.